# Portuguese validation of the Alcohol Craving Questionnaire–Short Form–Revised

**Rui Rodrigues, Eduardo López-Caneda, Natália Almeida-Antunes, Adriana Sampaio, Alberto Crego***

Psychological Neuroscience Laboratory (PNL), Research Center in Psychology (CIPSI), School of Psychology, University of Minho, Braga, Portugal

* alberto.crego@psi.uminho.pt

**Data Availability Statement:** All relevant data are within the manuscript and its Supporting Information files.

## Abstract

Alcohol craving has been described as a strong subjective desire to drink, being considered highly valuable in the clinical practice, as it is recognized as a strong predictor of alcohol relapse in alcohol-dependent individuals. However, to date, there is not a multifactorial questionnaire available for assessing short-term acute craving experience in Portugal. The aim of the present study was to validate a swift and efficient tool for the assessment of acute alcohol craving in a sample of Portuguese citizens. For that purpose, the Alcohol Craving Questionnaire–Short Form–Revised (ACQ-SF-R) was translated into European Portuguese and administered to a sample of 591 college participants with ages between 18 and 30 years. Results suggested that a three-factor model (i.e., Emotionality, Purposefulness, and Compulsivity) proved to be most suitable for the Portuguese sample. Overall, the ACQ-SF-R exhibited good psychometric properties, having a good internal consistency both for the general craving index (Cronbach's $\alpha = 0.85$) and each subscale (Cronbach's $\alpha = 0.66$–0.83), as well as an appropriate convergent validity with the Penn Alcohol Craving Scale ($r = 0.65$, $p < 0.001$), suggesting a good construct validity. In addition, the ACQ-SF-R also showed a good concurrent validity with the Alcohol Use Disorders Identification Test ($r = 0.57$, $p < 0.001$), indicating that risky alcohol use patterns are associated with increased craving scores in the ACQ-SF-R. Collectively, these findings suggest that the Portuguese version of the ACQ-SF-R can accurately measure alcohol craving at a multifactorial level, being a valid and reliable tool to use in Portuguese samples in research settings.

## Introduction

Alcohol craving has been classically described as a strong subjective desire to drink, being considered a hallmark of alcohol dependence [1, 2]. As such, it has been suggested that craving for alcohol is involved in the development and maintenance of severe alcohol use disorder (AUD), facilitating the emergence of alcohol-related obsessive thoughts and compulsive approach behaviors, despite repeated efforts to stop the consumption [3]. Therefore, craving conceptualization is considered highly valuable in clinical practice, as it is recognized as a strong predictor of alcohol relapse in alcohol-dependent individuals [4]. In addition, reduction

**Funding:** This study was conducted at the Psychology Research Centre (CIPsi/UM) School of Psychology, University of Minho, supported by the Foundation for Science and Technology (FCT) through the Portuguese State Budget (UIDB/ 01662/2020). The study was also supported by the project POCI-01-0145-FEDER-028672, funded by FCT and the European Regional Development Fund (FEDER). Eduardo López-Caneda and Alberto Crego were supported by the FCT and the Portuguese Ministry of Science, Technology and Higher Education, within the scope of the Individual Call to Scientific Employment Stimulus (CEECIND/ 02979/2018), and within the scope of the Transitory Disposition of the Decrete No. 57/2016, of 29th of August, amended by Law No. 57/ 2017 of 19 July, respectively. Natália Almeida-Antunes was supported by a fellowship from the FCT (SFRH/BD/146194/2019). https://ec.europa.eu/ regional_policy/pt/funding/erdf/ https://www.fct.pt/ The funders had no role in study design, data collection and analysis, decision to publish, or preparation of the manuscript.

**Competing interests:** The authors have declared that no competing interests exist.

of alcohol craving can lead to several benefits throughout rehabilitation, ultimately preventing relapses and boosting natural treatment outcomes [5], becoming one of the main targets for intervention in alcohol addiction [6, 7]. Indeed, the International Classification of Diseases (ICD-10 and ICD-11) [8, 9], as well as the fifth edition of the Diagnostic and Statistical Manual of Mental Disorders (DSM–5) [10] considered craving as a major criterion in the diagnosis of alcohol dependence and AUD, respectively.

The urge to drink or the intense and recurrent thoughts about alcohol, typically linked to alcohol craving, have often been associated with physiological and environmental cues [11]. Accordingly, specific environmental contexts (e.g., seeing/smelling alcohol) and physiological signals (i.e., bodily perceptions such as those linked to withdrawal symptoms) are known inducers of craving [3, 12–14]. Craving levels may also be time persistent, remaining low for long periods and still having high peak occurrences after weeks or months [11]. In this sense, it has been proposed that craving measures may be included within two main categories with respect to their timeframe: the state measures, which focus on the "here-and-now" craving status; and the global measures, which reflect the craving experience over the course of days, weeks or longer periods [1]. Additionally, recent investigation has described craving in terms of a triadic model, considering the craving experience as the interplay between three subcomponents: *cognitive craving*, associated with reduced efficiency in high-level cognitive abilities (e.g., inhibitory control); *automatic craving*, which represents the cue reactivity and the attentional bias to addiction-related cues; and *physiological craving*, expressed by body signals caused by a homeostatic imbalance such as stress or sleep deprivation [15].

The complexity of the craving phenomenon has led to the development of different questionnaires in order to capture the temporal nature of this experience as well as the potential sub-components or dimensions of craving. Specifically, two of the most commonly used questionnaires for assessing craving, namely the Alcohol Urge Questionnaire (AUQ) [16] and the Alcohol Craving Questionnaire (ACQ) [17], are measures of short-term acute craving experience. On the other hand, the Penn Alcohol Craving Scale (PACS) [18] and the Desires for Alcohol Questionnaire (DAQ) [19] retroactively assess past level of craving in the midterm. Concerning the dimensionality of these questionnaires, AUQ and PACS are characterized by having a unidimensional -single factor- structure, whereas ACQ and DAQ comprise multiple sub-scales or factors, thus being considered multidimensional.

Accordingly, as craving is often described as being composed of multiple subcomponents [1, 15], it is likely to be beneficial to use multifactorial scales for measuring craving, since these can have different predictive utility and offer contrasting treatment targets [20]. However, this type of scale usually has a great number of items, which has a negative effect on respondents' willingness to accurately complete them [21, 22]. Aiming to overcome these limitations, several short forms of multifactorial alcohol craving questionnaires have been developed and validated, including the six-item version of the DAQ (DAQ-6) [23] and the short form of the ACQ (ACQ-SF-R) [24]. Whereas DAQ-6 has six items divided into only two factors (i.e., "expectancy of negative reinforcement" and "strong desires and intentions to drink or use drugs"), the ACQ-SF-R is more comprehensive and comprises 12 items divided into four factors (i.e., Compulsivity, Expectancy, Purposefulness and Emotionality). This latter questionnaire has already been adapted and validated to the Brazilian and Spanish populations [25, 26], and it is considered a valuable and reliable measure widely used both in research and clinical settings to assess acute alcohol craving [27–29].

In Portugal, as far as we know, the PACS is the only alcohol craving measure that has been translated and validated to European Portuguese speakers [30]. However, this questionnaire simply allows assessing the craving levels retrospectively (during the past week) and as a single factor disregarding the individual's current craving and the potential multidimensional nature

of this phenomenon [15, 31]. With the intention of providing a swift and more efficient tool for the assessment of acute alcohol craving to researchers working with a sample of European Portuguese speakers, the objective of the present study was to develop and validate the Portuguese version of the ACQ-SF-R.

## Methods

### Participants

The initial sample included a total of 842 students from the University of Minho (Portugal). Data were collected between October 2018 and October 2019 by administering the questionnaires in a classroom setting. From the initial sample, 162 participants were excluded for not completing all questions of both questionnaires (ACQ-SF-R and PACS) and 89 for not meeting the age criteria (ages between 18–30 years old), which besides being Portuguese was the only inclusion criteria. There were no exclusion criteria. Thus, the final sample included 591 participants (67.3% female; see Table 1), of which eight did not complete the Alcohol Use Disorders Identification Test (AUDIT). Additionally, the sample was divided in two groups based on alcohol use pattern, i.e., non/low-risk drinkers (AUDIT < 8) and risky drinkers (AUDIT ≥ 8).

### Measures

*The Alcohol Craving Questionnaire–Short Form–Revised* (ACQ-SF-R) [24]. This questionnaire is a self-report measure consisting of 12 items derived from the 47-item Alcohol Craving Questionnaire (ACQ-Now) [17] and was designed to provide a brief screening method to be used in research and clinical practice to assess current craving for alcohol. Items are rated in a 7-point Likert-scale, from "totally disagree" (1) to "totally agree" (7), with the items 3, 8, and 11 being inversely scored, and a general craving index may be derived by summing all items and dividing by 12.

According to Singleton [24], the factor analysis of the ACQ-SF-R supported the same four factors as the ACQ-Now, with a good internal consistency (Cronbach's $\alpha$ ranged between 0.77 and 0.86). These factors measure different dimensions of craving for alcohol: *Compulsivity*, related to the loss of control over drinking; *Expectancy*, concerned with the positive benefits of drinking; *Purposefulness*, associated with the intention and plan to drink; and *Emotionality*, linked to the relief from withdrawal/negative affect [32].

*Penn Alcohol Craving Scale* (PACS) [18]. The PACS is a self-report measure with five questions, each one with seven answer options (from 0 to 6). The first three questions range

**Table 1. Age and alcohol consumption by gender and alcohol use pattern.**

|  | Age *M(SD)* | AUDIT *M(SD)* | PACS *M(SD)* | ACQ-SF-R *M(SD)* |
|---|---|---|---|---|
| **Gender** |  |  |  |  |
| Men (N = 193) | 20.65 (2.16) | 4.68 (4.35) | 2.40 (3.45) | 2.18 (1.00) |
| Women (N = 398) | 20.23 (1.64) | 5.11 (5.47) | 1.73 (3.35) | 1.80 (0.86) |
| **Alcohol Use Pattern** |  |  |  |  |
| Risky Drinkers (N = 129) | 20.29 (1.60) | 12.88 (5.47) | 5.08 (5.04) | 2.73(1.10) |
| Non/low Risk Drinkers (N = 454) | 20.43 (1.89) | 2.59 (2.20) | 1.07 (2.06) | 1.70(0.73) |
| Total Sample (N = 591) | 20.37 (1.84) | 4.97 (5.13) | 1.95 (3.40) | 1.93(0.93) |

Note: M, mean; SD, standard deviation; AUDIT, Alcohol Use Disorders Identification Test; PACS, Penn Alcohol Craving Scale; ACQ-SF-R, Alcohol Craving Questionnaire–Short Form–Revised.

from different levels of intensity, frequency and duration of alcohol-related thoughts. The fourth question concerns the alcohol consumption restraint capacity and the last one focuses on the levels of craving in the past week. This scale has been translated and valitated for European Portuguese speakers, and presents good psychometric properties [30].

***Alcohol Use Disorders Identification Test* (AUDIT) [33].** The AUDIT is a self-report measure with ten questions, used to identify current harmful and hazardous drinking. This test has been validated to assess alcohol-related problems and/or AUD [34] and has become the world's most widely used alcohol screening instrument, including in Portugal [35]. AUDIT scores < 8 reveal low risk of alcohol use; scores between 8 and 15 represent a risky consumption; scores from 16 to 19 are considered a harmful intake pattern; and scores ≥ 20 indicate very high risk for alcohol dependence and warrant further diagnostic evaluation for alcohol dependence [33, 36, 37]. In the present study, we used the European Portuguese translated version of the AUDIT [37].

## Procedure

The study consisted of three phases: (1) scale translation; (2) questionnaire administration; and (3) analysis of validity and reliability.

In the first phase, ACQ-SF-R was independently translated from the English version into European Portuguese by four Portuguese native researchers with a high level of proficiency in English and, subsequently, compared and discussed in a small group of expert researchers in the topic. This version was then back translated into English by an independent researcher with English as first language and compared with the original English questionnaire. After verifying the equality of contents between the original and the final translation as well as the proper understanding of all items, the final version was created (see S1 Appendix).

In the second phase, the two alcohol craving measurements (ACQ-SF-R and PACS) were administered, along with the AUDIT, to college students at University of Minho in a classroom context, with the professor and students' verbal consent. The Ethics Subcommittee for Social and Human Sciences (SECSH) of the University of Minho gave written approval for this study, associated with the approval number CE.CSH 078/2018.

During the third phase, statistical analyses of validation and reliability of the Portuguese version of ACQ-SF-R were performed. The guidelines of the General Data Protection Regulation (UE 2016/679) and the Declaration of Helsinki [38] were followed to guarantee the confidential treatment of the data and to ensure compliance with international standards of ethical research involving human beings.

## Data analysis

Firstly, an Exploratory Factorial Analysis (EFA) with a Varimax rotation was conducted to identify the factorial structure of the scale. The method of extraction used was unweighted least squares (ULS), and factors with eigenvalue below one were excluded. Kaiser Meyer Olkin (KMO) and Bartlett's Test were performed as a preliminary step to determine the properties of the inter-item correlation matrix and to measure the strength of the relationship between factors. The items were checked for double loading to a factor (i.e., having a loading > 0.60 to more than one factor). Subsequently, a Confirmatory Factor Analysis was completed by the Maximum Likelihood method to test the Portuguese three-factor model and the original four-factor model. To this extent, we considered the following fit indexes: Relative Chi-Square ($x2/df$); Normed Fit Index (NFI); Comparative Fit Index (CFI); Goodness-of-Fit Index (GFI); and Root Mean Square Error of Approximation (RMSEA). In accordance, commonly accepted fit indices and acceptable limits for model fit are presented in Table 2 [39, 40]. Akaike

**Table 2. Goodness of fit indices and acceptable limits.**

| Indices | Acceptable limits |
|---------|-------------------|
| $\chi 2/df$ | $\leq 5$ acceptable fit, $\leq 2$ perfect fit |
| RMSEA | 0.10–0.08 weak fit, $\leq 0.08$ good fit, $\leq 0.06$ perfect fit |
| GFI | 0.85–0.89 acceptable fit, $\geq 0.90$ good fit, $\geq 0.95$ perfect fit |
| CFI | $\geq 0.90$ acceptable fit, $\geq 0.95$ good fit, $\geq 0.97$ perfect fit |
| NFI | $\geq 0.90$ acceptable fit, $\geq 0.95$ good fit |

Note: $\chi 2$, chi-Square; *df*, degrees of freedom; RMSEA, Root mean square error of approximation; GFI, Goodness-of-fit statistic; CFI, Comparative fit index; NFI, Normed-fit index.

information criterion (AIC) was also considered for the comparisons between different models, where smaller values represent the best model.

The internal consistency of the full scale and each factor was calculated using the Cronbach's $\alpha$ coefficient and the mean inter-item correlation was calculated to confirm if each individual item provides a consistent and appropriate measurement of the construct. Furthermore, Pearson's correlations were conducted to assess convergent validity using PACS data as reference, and concurrent validity using AUDIT scores. Additionally, in order to compare ACQ-SF-R scores for participants in the two levels alcohol risk (see Table 1), a repeated measures ANOVA was conducted with one between-subjects factor with two levels (alcohol use pattern: risk and non/low risk) and one within-subjects factor with three levels (ACQ-SF-R: Emotionality, Purposefulness and Compulsivity) to evaluate potential differences in the craving levels between these two groups.

Finally, configurational invariance across gender was also explored. Specifically, EFA, KMO and Bartlett's Test, as well as the Cronbach's $\alpha$ coefficient and mean inter-item correlation, were conducted separately within each gender.

## Results

### Exploratory Factor Analysis (EFA)

Results of the Kaiser-Meyer-Olkin measure of sampling adequacy and Bartlett's test of sphericity were appropriate for factor analysis (KMO = 0.87; $\chi 2$ = 2429.56; $p < 0.001$). Three different factors, which together accounted for 60.64% of the total variance were extracted (see Table 3).

**Table 3. Total explained variance from the exploratory factor analysis.**

| Factor | Initial Eigenvalues | | | Extraction Sums of Squared Loading | | | Rotation Sums of Squared Loadings | | |
|--------|-------|------------|--------------|-------|------------|--------------|-------|------------|--------------|
| | Total | Variance % | Cumulative % | Total | Variance % | Cumulative % | Total | Variance % | Cumulative % |
| 1 | 4.628 | 38.567 | 38.567 | 4.628 | 38.567 | 38.567 | 3.037 | 25.312 | 25.312 |
| 2 | 1.453 | 12.107 | 50.674 | 1.453 | 12.107 | 50.674 | 2.381 | 19.842 | 45.153 |
| 3 | 1.201 | 10.011 | 60.685 | 1.201 | 10.011 | 60.685 | 1.864 | 15.532 | 60.685 |
| 4 | 0.762 | 6.350 | 67.035 | | | | | | |
| 5 | 0.710 | 5.913 | 72.948 | | | | | | |
| 6 | 0.624 | 5.198 | 78.146 | | | | | | |
| 7 | 0.585 | 4.879 | 83.025 | | | | | | |
| 8 | 0.538 | 4.487 | 87.512 | | | | | | |
| 9 | 0.456 | 3.803 | 91.315 | | | | | | |
| 10 | 0.413 | 3.444 | 94.759 | | | | | | |
| 11 | 0.360 | 3.001 | 97.760 | | | | | | |
| 12 | 0.269 | 2.240 | 100.000 | | | | | | |

Factor 1, Emotionality, explained 38.58% of the variance and it was comprised of items 6, 7, 9, 10 and 12. Factor 2, Purposefulness, explained 12.03% of the variance and it included the items 3, 8 and 11. Factor 3, Compulsivity, explained 10.03% of the variance and encompassed items 1, 2, 4 and 5. None of the items were double loaded on any factor.

### Confirmatory factor analysis

In conformity with the Maximum Likelihood method, the three-factor model and the original four-factor model were tested. The three-factor model reached the following fit indices: $x2/df$ = 4.98, NFI = 0.90, CFI = 0.92, GFI = .94, RMSEA = 0.08, while the four-factor model reached the following fit indices: $x2/df$ = 6.50, NFI = 0.87, CFI = 0.89, GFI = 0.91, RMSEA = 0.10. Each model scored 307.72 and 372.14 for AIC, respectively. The upper mentioned indices and AIC indicated a better model fit for the three-factor model (see Fig 1 and, Tables 2 and 4).

### Reliability

Both the total scale ($\alpha$ = 0.85) and the three factors (F1, $\alpha$ = 0.83; F2, $\alpha$ = 0.66; F3, $\alpha$ = 0.70) showed good internal consistency values [41]. Calculation of the mean inter-item correlation also showed good internal consistency for the full scale (0.32) and the three factors (F1 = 0.32; F2 = 0.40; F3 = 0.37) [42]. Finally, the reliability of the scale did not improve by removing any of the items.

### Validity

A significant correlation was found when PACS was compared with the full-scale score of the ACQ-SF-R ($r$ = 0.65, $p < 0.001$), indicating good convergent validity. Furthermore, a significant correlation was also observed between ACQ-SF-R and AUDIT ($r$ = 0.57, $p < 0.001$), indicative of appropriate concurrent validity. Additionally, comparisons between risky drinkers and non/low risk drinkers showed significant differences regarding their craving scores ($F$ (1, 581) = 155.86, $p < 0.001$). Specifically, risky drinkers reported higher scores of alcohol craving for all three subscales in comparison with non/low risk drinkers ($F$ (2, 1655 = 16.91, $p < 0.001$) (Table 5).

### Factorial invariance and reliability across gender

When conducting EFAs separately within each gender, results of both female (KMO = .84; $\chi 2$ = 1638.29, $p < 0.001$) and male groups (KMO = 0.84; $\chi 2$ = 799.69, $p < 0.001$) were appropriate. EFA showed that males and females had identical factor loadings in all the subscales as the main exploratory analysis, except for item 1 which had a higher factor loading on F2 (Purposefulness) than F3 (Compulsivity) for females. Additionally, the three factors explained similar amounts of variance, namely F1, F2 and F3 explained 38.49%, 12.49% and 9.50% of the variance for males, and 37.42%, 12.93% and 9.92% for females.

Furthermore, good internal consistency values in the total scale and the three subscales were expressed for both males (Total scale: $\alpha$ = 0.84; F1: $\alpha$ = 0.81; F2: $\alpha$ = 0.60; F3: $\alpha$ = 0.72) and females (Total scale: $\alpha$ = 0.83; F1: $\alpha$ = 0.81; F2: $\alpha$ = 0.69; F3: $\alpha$ = 0.64). Likewise, mean inter-item correlations also showed good internal consistency in the full scale and the three factors, in both male (Total scale: $\alpha$ = 0.31; F1: $\alpha$ = 0.49; F2: $\alpha$ = 0.33; F3: $\alpha$ = 0.42) and female (Total scale: $\alpha$ = 0.30; F1: $\alpha$ = 0.50; F2: $\alpha$ = 0.42; F3: $\alpha$ = 0.32) groups.

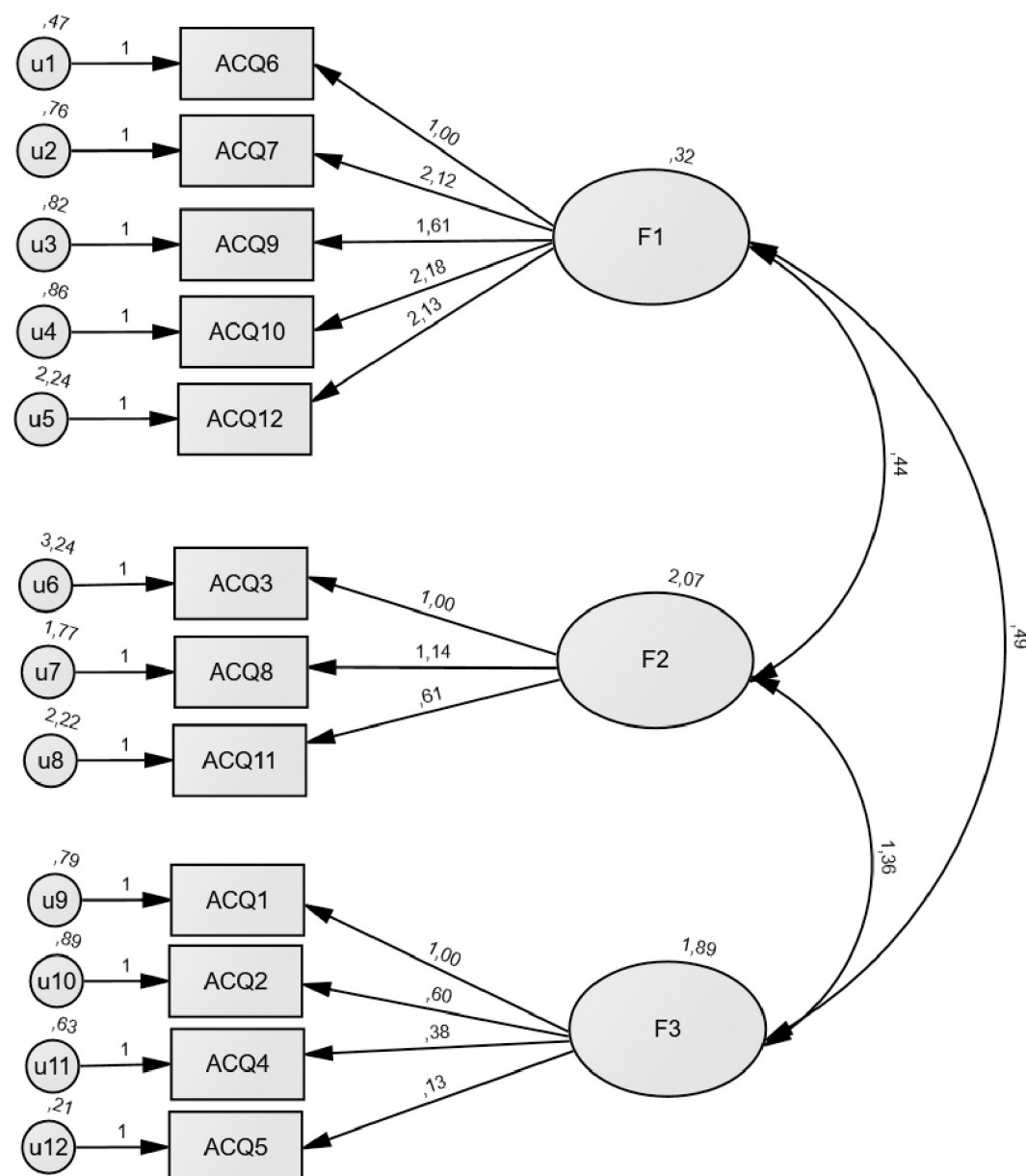

**Fig 1. Confirmatory factor analysis for the ACQ-SR-N-PT F1 -Emotionality- includes items 6, 7, 9, 10, and 12; F2 -Purposefulness- includes items 3, 8, and 11; F3 –Compulsivity includes items 1, 2, 4, and 5 (N = 591).**

**Table 4. Model fit indices.**

| Model | $\chi^2$ | df | $\chi^2$/df | NFI | CFI | GFI | RMSEA | AIC |
|---|---|---|---|---|---|---|---|---|
| Three-factor | 253.72 | 51 | 4.98 | 0.90 | 0.92 | 0.94 | 0.08 | 307.72 |
| Four-factor | 312.14 | 48 | 6.50 | 0.87 | 0.89 | 0.91 | 0.10 | 372.14 |

Note: $\chi^2$ –Chi-Square; *df*–degrees of freedom; NFI–Normed Fit Index; CFI–Comparative Fit Index; GFI–Goodness-of-Fit Index; RMSEA–Root Mean Square Error of Approximation; AIC–Akaike Information Criterion.

**Table 5. Craving scores across alcohol use pattern groups.**

| | Non/Low Risk Drinkers M(SD) | Risky Drinkers M(SD) |
|---|---|---|
| | N = 454 | N = 129 |
| F1—Emotionality | 1.60 (0.91) | 2.56 (1.40) |
| F2—Purposefulness | 2.35 (1.45) | 3.83 (1.57) |
| F3—Compulsivity | 1.34 (0.58) | 2.11 (1.21) |
| ACQ-SF-R–Full Scale | 1.70 (0.73) | 2.72 (1.10) |

Note: ACQ-SF-R, Alcohol Craving Questionnaire–Short Form–Revised; M, mean; SD, standard deviation.

## Discussion

The ACQ-SF-R is a widely used screening instrument to assess craving in both clinical and non-clinical populations. However, there are no studies to date assessing the psychometric properties of this questionnaire in Portugal. Thus, the main objective of the present study was to adapt and validate a version of the ACQ-SF-R for European Portuguese speakers. The findings of the present study suggest that the ACQ-SF-R possesses good psychometric properties and is a valid and reliable measure to assess acute alcohol craving in three complementary dimensions: Emotionality, Purposefulness, and Compulsivity. Additionally, the instrument showed good convergent and concurrent validity and showed cohesiveness within each gender, with good internal consistency of the scale for both males and females.

Regarding instrument reliability, results revealed good internal consistency values ($\alpha$ = 0.85), with similar values to the original version of the ACQ-SF-R ($\alpha$ range between 0.77 and 0.86; Singleton, 1997) as well as the Spanish ($\alpha$ = 0.87; Gálvez et al., 2016) and the Brazilian ($\alpha$ = 0.91; Girelli et al., 2019) validations. Similarly, the Portuguese version of the ACQ-SF-R showed good convergent validity with the PACS, with a higher relationship strength than the Spanish ACQ-SF-R validation [25]. Thus, the present study seems to complement the psychometric results found for the Portuguese version of the PACS, which to date is the only questionnaire assessing alcohol craving validated in European Portuguese. Furthermore, and contrary to the PACS, ACQ-SF-R also allows the assessment of different dimensions of alcohol craving, which can be useful to differentiate how this phenomenon is expressed in different populations (e.g., light drinkers, binge drinkers, alcohol dependent individuals).

Additionally, Pearson's correlations showed satisfactory concurrent validity between the ACQ-SF-R and the AUDIT ($r$ = 0.57), reflecting an interaction between alcohol craving and alcohol consumption. In a related vein, levels of alcohol craving were significantly higher in risky drinkers when compared to non/low risk drinkers. Indeed, alcohol craving has often been associated with higher detrimental patterns of alcohol use [3, 43, 44]. Consequently, ACQ-SF-R may be a useful tool for clinical practice, being able to help to detect risky drinking, and future research could determine whether this questionnaire will be useful in predicting risk for relapse and other treatment outcomes.

According to Singleton [24], the English version of ACQ-SF-R is a four-factor measure, however, in our study, the confirmatory factor analysis showed better fit indices for the three-factor model when compared to the original model. Nonetheless, with the number of factors reduced to three, the Portuguese version of the ACQ-SF-R still presented a good construct validity [45] with a total explained variance of 60.6%. Concerning the overall structure of the scale, in the Portuguese version the Expectancy factor was removed, and its items merged into the Emotionality and Compulsivity factors. Furthermore, item 6, which previously loaded on the Compulsivity factor, was included in the Emotionality factor. Thus, the Emotionality dimension included five items (i.e., 6, 7, 9, 10, 12) that paired abstinence from alcohol with

feelings of irritability, tension, and restlessness together with the anticipation of some benefits of drinking (e.g., on mood); the Compulsivity factor was composed of four items (i.e., 1, 2, 4, 5) reflecting the urges and desires to drink along with the ability to refrain from consuming alcohol; and, finally, the Purposefulness factor remained identical to the original scale, characterized by three items (i.e., 3, 8, 11) reflecting the intention and plan to drink alcohol.

Consistent with the present study, both the Spanish and Brazilian validations [25, 26] suggested better fit with a smaller number of factors than the original version of the ACQ-SF-R, i.e., three factors for the Spanish validation and two factors for the Brazilian one. With regard to the latter, there are a number of differences between the Brazilian version and the current validation that are worth mentioning. Participants of the Brazilian study were patients recently admitted into a detoxification facility, therefore they had high motivation to change [46] and, consequently, scored low on the scale of purposefulness in regard to drinking. In addition, prior to the craving assessment, these patients were exposed to two pictures of alcoholic beverages, which could lead to incremental changes in craving levels [14] and might have increased both the positive alcohol expectancies and the compulsive tendencies to approach alcohol [47, 48]. Thus, it could be suggested that this previous exposure to alcohol cues may have led to this co-occurrence, causing items associated with expectancies, compulsivity, and emotionality to have higher correlation, which may entail a bigger loading to the same factor [49]. Therefore, these methodological differences might have led to the structural differentiation between the Brazilian and the Portuguese validations.

Finally, the present study has some limitations that deserve consideration. While a strength of the study was that the sample size was higher than previous validation studies of the scale [25, 26], the nature of the sample (young college students) may limit the generalization of the results to the adult population or population without higher education. In the same vein, although part of the sample had a risky consumption pattern, participants cannot be considered a clinical population. As such, results from this validation should not be generalized to clinical populations (e.g., patients with AUD). Moreover, only students from a single public university completed the questionnaire, hence external validity concerning the sociodemographic characteristics of the population might be limited. In addition, the fact that this study presents a significantly higher number of females than males (67.3% vs 32.7%), might be affecting our results. Lastly, all variables analyzed were self-reported, which makes it difficult to determine possible underestimation/overestimation of the levels of alcohol consumption. Thus, additional studies trying to equate the number of each gender and include a general adult population should be conducted in order to validate the present results.

In conclusion, the Portuguese version of the ACQ-SF-R showed good psychometric properties, i.e., revealing good internal consistency and satisfactory convergent validity with the PACS and concurrent validity with the AUDIT. Furthermore, despite having a shorter length, it allows assessment of different dimensions of alcohol craving (i.e., Emotionality, Purposefulness and Compulsivity). Collectively, these findings suggest that this version of the ACQ-SF-R can accurately measure acute alcohol craving at a multifactorial level, being a valid and reliable tool to use in Portuguese samples in research settings.

## Supporting information

**S1 Appendix. Administration sheet of the Portuguese version of the Alcohol Craving Questionnaire-Short Form-Revised.**
(PDF)

**S1 Data. File concerning all data analysed in this article.**
(SAV)

## Acknowledgments

The authors thank Dr. Joana Coutinho and Dr. Sónia S. Sousa for their help as independent translator of the scale for Portuguese as well as Dr. Harry Moore for his contribution in the back-translation of the scale into English.

## Author Contributions

**Conceptualization:** Rui Rodrigues, Eduardo López-Caneda, Natália Almeida-Antunes, Alberto Crego.

**Data curation:** Rui Rodrigues, Eduardo López-Caneda, Natália Almeida-Antunes, Alberto Crego.

**Formal analysis:** Eduardo López-Caneda, Alberto Crego.

**Funding acquisition:** Eduardo López-Caneda, Natália Almeida-Antunes, Alberto Crego.

**Investigation:** Rui Rodrigues, Eduardo López-Caneda, Natália Almeida-Antunes, Alberto Crego.

**Methodology:** Rui Rodrigues, Eduardo López-Caneda, Natália Almeida-Antunes, Alberto Crego.

**Project administration:** Eduardo López-Caneda, Adriana Sampaio, Alberto Crego.

**Supervision:** Eduardo López-Caneda, Alberto Crego.

**Validation:** Eduardo López-Caneda, Adriana Sampaio, Alberto Crego.

**Visualization:** Rui Rodrigues.

**Writing – original draft:** Rui Rodrigues.

**Writing – review & editing:** Rui Rodrigues, Eduardo López-Caneda, Natália Almeida-Antunes, Adriana Sampaio, Alberto Crego.

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
