## [Decision Letter · Decision Letter 0]

9 Mar 2021

PONE-D-21-05321

Portuguese Validation of the Alcohol Craving Questionnaire – Short Form – Revised

PLOS ONE

Dear Dr. Crego,

Thank you for submitting your manuscript to PLOS ONE. After careful consideration, we feel that it has merit but does not fully meet PLOS ONE’s publication criteria as it currently stands. Therefore, we invite you to submit a revised version of the manuscript that addresses the points raised during the review process.

We look forward to receiving your revised manuscript.

Kind regards,

Wen-Jun Tu

Academic Editor

PLOS ONE

Journal Requirements:

3. PLOS ONE has specific requirements for studies that are presenting a new method or tool as the primary focus, including a newly developed or modified questionnaire or scale (https://journals.plos.org/plosone/s/submission-guidelines#loc-methods-software-databases-and-tools. To that effect, in your Methods section, please discuss whether you obtained the necessary permissions from the owner of the original questionnaire to modify it.

Reviewers' comments:

Reviewer's Responses to Questions

**Comments to the Author**

1. Is the manuscript technically sound, and do the data support the conclusions?

Reviewer #1: Yes

Reviewer #2: Yes

2. Has the statistical analysis been performed appropriately and rigorously? 

Reviewer #1: Yes

Reviewer #2: Yes

3. Have the authors made all data underlying the findings in their manuscript fully available?

Reviewer #1: Yes

Reviewer #2: Yes

4. Is the manuscript presented in an intelligible fashion and written in standard English?

Reviewer #1: No

Reviewer #2: Yes

5. Review Comments to the Author

Reviewer #1: Writing corrections:

Does PLOS-One use the American or European system of decimals (page 5 shows “67,3%” where the comma should be a decimal). Page 5: “8” should be “eight” under standard professional rules for use of numerals. “To this extend” should be “To this extent” (page 8). On page 12, change “for a by gender factor analysis” to “for a factor analysis by gender”, and change “similar variances” to “similar amounts of variance”. Bottom of p 13: “this scale also allows to evaluate” should be “this scale also allows one to evaluate”. Middle of p. 14: change “which previously was comprised in the Compulsivity dimension,” to “which previously loaded on the Compulsivity factor”. Change “Compulsivity factor was composed by” to “Compulsivity factor was composed of”. Change “ability to restrain from” to “ability to refrain from”. Page 15 (top): a version does not have participants – a study does, so change “version” to “study” in the 4th line, then change “who have” to “so they had” (it was not a sentence in the way it was written, and past tense must be used). Remove “As such” (not appropriate in this sentence). Change “changes on craving levels” to “changes in craving levels”. Bottom of p 15: Incorrect use of the term “fulfilled”: change “fulfilled” to “completed” or “filled out”. Top of page 16: change “Concluding” to “In conclusion” for correct grammar. The phrase “having a short extension” has no meaning”; reword that. What was meant by “extension”?

Other comments that need addressing:

Introduction and Discussion: Was the Brazilian version of the ACQ-SF-R also in Portuguese, given that Portuguese is the national language of Brazil and that their manuscript is in Portuguese? Would it therefore make it untrue that this manuscript’s version is the first one in Portuguese? Is it instead true that this is the first one in European Portuguese? The manuscript needs to be accurate about this. Maybe sometimes the word “Portuguese” is being used to refer to the country studied rather than the language, but this is highly confusing, if so, since Portuguese is the language of both Portugal and Brazil.

Discussion: Bottom of p. 13: It is inaccurate to say this translation of the measure was given to a clinical population since the participants were college students not recruited from a clinical setting. The fact that many are risky drinkers does not qualify this as clinical. Remove any statements about this translation of the measure being validated in a clinical population. The following sentence is confusing: “This present study seems to complement the only Portuguese-validated questionnaire so far assessing craving in clinical and non-clinical population”. Does this mean the present study complements the Brazilian study’s Portuguese validated questionnaire results? Is the word “complement” a mistranslation of some other intended meaning? A study cannot complement itself as the sentence seems to be saying.

Method: State in the Measures section which of these measures have previously been translated and validated in Portuguese.

Say if the “consent” was written informed consent or oral informed consent. (page 8)

Say what method was used to identify number of factors in the exploratory step (it looks like the Eigenvalue method was used rather than Minimum Partial or Parallel Analysis), or say that the analysis was constrained to three or to four factors. That should be done before the steps that follow the first sentence.

Before doing the steps on page 9, it is usual to check to see if any items are double loaded (e.g., load > .60 on a second factor) and remove them from the final factors.

Combine the two validity sections under one header, called “Validity, with no sub-headers, and make it one continuous paragraph. (A one sentence paragraph is too short to stand alone).

Abstract: Add the word “college” before “participants” or “in college” after “participants”.

Reviewer #2: This manuscript reports the results of a study validating the Alcohol Craving Questionnaire – Short Form - Revised (ACQ-SF-R) in a sample of European Portuguese speakers.

The study was well-designed in its methods, with some limitations, and included Portuguese-speaking college students. The AUDIT was used to classify individuals as “risky drinkers” or non/low risk drinkers.” The ACQ-SF-R indicated that “risky drinkers” had increased craving scores on the ACQ-SF-R. Overall, the ACQ-SF-R was found to have good validity and reliability in the study population.

The manuscript was easy to follow and the data were statistically accurately analyzed. The results support the need to continue to study this instrument in a more representative sample. Some format f the language construction has a very European style; I would recommend to have an English native speaker to review the manuscript prior resubmission.

Below are some specific comments to improve the manuscript:

Abstract

1) Please change “administrated” to “administered.”

Introduction

1) Please change “However, this type of scales is usually” to “However, this type of scale is usually” or “However, these types of scales are usually.”

Methods

1) Please change the comma on page 5 to a decimal: “67,3%” to “67.3%.”

2) I would suggest stating a bit earlier in the Methods that the AUDIT was used to classify individuals as either “risky drinkers” or “non/low risk drinkers” since Table 1 appears pages before that explanation.

3) In the Data Analysis subsection, please change “To this extend” to “To this extent.”

Results

1) Please italicize all statistical symbols.

2) Please add a zero in front of decimals: .87 becomes 0.87, etc.

3) Change sentences beginning with “Significant correlation” to “A significant correlation.”

Discussion

1) On page 13, please change “contrarily” to “contrary.”

2) On page 15 where study limitations are discussed, age and income are mentioned, but the disproportionate number of women (n = 398) to men (n = 193) in the study is not discussed. Please include this limitation in the Discussion section on page 15.

6. PLOS authors have the option to publish the peer review history of their article (what does this mean?). If published, this will include your full peer review and any attached files.

Reviewer #1: No

Reviewer #2: No

---

## [Author Response · Author response to Decision Letter 0]

17 Mar 2021

We thank the reviewers for their useful suggestions. In the next pages, we try to address each of their considerations.

Reviewer 1

First of all, we thank the reviewer for their valuable comments and suggestions as we believe they have helped substantially to improve the manuscript.

Issue 1. Writing corrections

1. In the abstract, “college” was added before “participants” for a better understanding and more specific characterization of the sample. 

2. On page 5 where it showed “67,3%” has been changed to “67.3%”, in accordance with the correct system of decimals. All other decimals have been checked.

3. Also, on page 5, “8” has been changed to “eight”, to meet the standard professional rules for use of numerals. Additionally, the manuscript was checked for similar cases.

4. On page 8, “to this extend” has been changed to “to this extent”.

5. On page 12, “for a by gender factor analysis” has been changed to “for a factor analysis by gender” and “similar variances” has been changed to “similar amounts of variance”.

6. On page 13, “this scale also allows to evaluate” has been changed to “this scale also allows one to evaluate”. 

7. On page 14, “which previously was comprised in the Compulsivity dimension,” has been changed to “which previously loaded on the Compulsivity factor”; “Compulsivity factor was composed by” has been changed to “Compulsivity factor was composed of”; “ability to restrain from” has been changed to “ability to refrain from”. 

8. On page 15, “version” has been changed to “study” and “who have” has been changed to “so they had”; “As such” was removed from the sentence; “changes on craving levels” has been changed to “changes in craving levels”; “fulfilled” has been changed to “completed”. 

9. On page 16, “Concluding” has been changed to “In conclusion”.

10. The phrase “having a short extension” was reworked and changed to “having a shorter length”.

Additionally, the manuscript was revised by an independent person with high proficiency in English and further changes were made regarding grammar.

Issue 2. Clarification concerning the use of European or Brazilian Portuguese when referring to the sample and validation language

The authors would like to thank the Reviewer 1 for highlighting these issues since it could, in fact, lead the reader towards misconceptions regarding the nature of the study and its objectives. In this sense, the manuscript was fully reviewed, and expressions “Portuguese citizens” and “European Portuguese speakers” were introduced to provide more clarity. 

Despite it remains true that the Brazilian validation of ACQ-SF-R was developed in Portuguese, there are differences between Brazilian Portuguese and European Portuguese. Additionally, the need for a validation in European Portuguese was also tied in with the sociocultural differences between Brazilian and Portuguese citizens. In this sense, this is, indeed, the first validation of the scale for the population of Portugal, while also being the first in European Portuguese.

Issue 3. Inaccurate statements suggesting the scale validation in a clinical sample

We thank the reviewer for bringing our attention to these inaccurate statements. All mentions of the use of a clinical sample, or possible validation of the scale in a clinical population have been removed from the manuscript. Additionally, in the end of the discussion (pp. 16, lines 325 and 335) we suggested that future investigation should include a clinical sample aiming to strengthen the scale’s generalization properties, while also enabling its use in these populations.

“Although part of the sample had a risky consumption pattern, participants cannot be considered a clinical population. As such, results from this validation should not be generalized to clinical populations (e.g., patients with AUD)”.

“Further research including a greater sociodemographic representation as well as the inclusion of a clinical (i.e., alcohol-dependent) group is recommended in order to enhance the external strength and generalization properties of the questionnaire (e.g., for using in clinical setting).”.

Issue 4. Confusing sentence regarding ACQ-SF-R being a complement to PACS

As it was highlighted by reviewer, the sentence “The present study seems to complement the only Portuguese-validated questionnaire so far assessing craving in clinical and non-clinical population” was indeed confusing. In this sense, the full sentence was amended to “Thus, the present study seems to complement the PACS, which to date is the only questionnaire validated in European Portuguese assessing alcohol craving” providing a better understanding of its true meaning. In this sense, what the sentence should imply is that ACQ-SF-R comes to fill possible gaps that PACS had (e.g., only having a single factor), and together they can mutually enhance each other.

Nevertheless, if the use of the word “complement” remains confusing or misused we will gladly change it for another that fits better in the meaning of the sentence.

Issue 5. Missing information in the Method

In agreement with reviewer’s suggestions, PACS has now been clearly described as being a measure previously validated to the Portuguese population. To our knowledge, there is no validation of the AUDIT questionnaire to the Portuguese population, notwithstanding we followed the translation and the values suggested by Serviço de Intervenção nos Comportamentos Aditivos e nas Dependências (2013), which is one of the most regarded institutions in the field of substance abuse in Portugal. Additionally, information specifying the use of verbal informed consent was added. 

As described by reviewer 1, the method used to identify the number of factors in the exploratory step was, indeed, Eigenvalue (i.e., a factor was considered when its Eigenvalue was higher than 1), and this information has now been described in the manuscript. Likewise, we followed the reviewer’s advice and performed a Parallel Analysis. Results from this analysis were identical to those obtained with Eigenvalue (see Table 1), in a sense that the same number factors met the selection criteria (i.e., having an Eigenvalue in the Exploratory Factor Analysis superior to the random generated one). We decided not to include the Parallel Analysis in the final version of the manuscript due to a possible redundance with the Eigenvalue approach. However, we highly value the reviewer’s suggestion and would gladly make further changes on this matter if the reviewer deemed necessary. 

 Minor Issues.

As suggested by the reviewer, items were checked for double loading and it was accordingly described in the new version of the manuscript (pp. 8, line 172): “The items were checked for double loading to a factor (i.e., having a loading > 0.60 to more than one factor)”. Also, as suggested by the reviewer, the two validity sections were combined under a single header called “Validity”.

Reviewer 2

We thank the reviewer for the overall positive evaluation of the manuscript and for their meaningful comments, that we have carefully tried to address below. Additionally, and as suggested, the whole manuscript was revised by an independent person with a high degree of proficiency in English.

Issue 1. Writing corrections

1. In the Abstract, “administrated” has been changed to “administered”.

2. In the Introduction, “However, this type of scales is usually” has been changed to “However, this type of scale is usually”.

3. On page 5 where it showed “67,3%” has been changed to “67.3%”, in accordance with the correct system of decimals. All other decimals have been checked. A zero has been added in front of all decimals.

4. In Data Analysis, “To this extend” has been changed to “To this extent”.

5. In the Results, sentences beginning with “Significant correlation” have been changed to “A significant correlation”.

6. In the Discussion, “contrarily” has been changed to “contrary”.

Issue 2. Missing description of “risky drinkers” and “non/low risk drinkers” prior to Table 1

As suggested, it has been stated earlier in the Methods section (pp. 6, line 111) the classification of individuals regarding their pattern of alcohol use. Thus, the meaning of “risky drinkers” and “non/low risk drinkers” has been explicitly described prior to Table 1 appearance.

In this sense the following was added to the manuscript “Additionally, the sample was divided in two groups based on alcohol use pattern, i.e., non/low drinkers (AUDIT < 8) and risky drinkers (AUDIT ≥ 8).” 

Issue 3. Limitation regarding disproportionate number of women to men was not discussed

As pointed out by Reviewer 2, this limitation is, indeed, important and should be thoroughly discussed. As such, we added information stating that “In addition, the fact that this study presents a significantly higher number of females than males (67.3% vs 32.7%), might be affecting our results. Thus, additional studies trying to match the gender distribution should be conducted in order to validate the present results.”

References

Serviço de Intervenção nos Comportamentos Aditivos e nas Dependências. AUDIT - Alcohol Use Disorders Identification Test. Serviço de Intervenção Nos Comportamentos Aditivos e Nas Depências. 2013. Available from: http://www.sicad.pt/PT/Intervencao/RedeReferenciacao/SitePages/detalhe.aspx?itemId=2&lista=SICAD_INSTRUMENTOS&bkUrl=/BK/Intervencao/RedeReferenciacao/

---

## [Decision Letter · Decision Letter 1]

13 Apr 2021

PONE-D-21-05321R1

Portuguese Validation of the Alcohol Craving Questionnaire – Short Form – Revised

PLOS ONE

Dear Dr. Crego,

Thank you for submitting your manuscript to PLOS ONE. After careful consideration, we feel that it has merit but does not fully meet PLOS ONE’s publication criteria as it currently stands. Therefore, we invite you to submit a revised version of the manuscript that addresses the points raised during the review process.

We look forward to receiving your revised manuscript.

Kind regards,

Wen-Jun Tu

Academic Editor

PLOS ONE

Journal Requirements:

Reviewers' comments:

Reviewer's Responses to Questions

**Comments to the Author**

1. If the authors have adequately addressed your comments raised in a previous round of review and you feel that this manuscript is now acceptable for publication, you may indicate that here to bypass the “Comments to the Author” section, enter your conflict of interest statement in the “Confidential to Editor” section, and submit your "Accept" recommendation.

Reviewer #1: (No Response)

Reviewer #2: All comments have been addressed

2. Is the manuscript technically sound, and do the data support the conclusions?

Reviewer #1: Yes

Reviewer #2: Yes

3. Has the statistical analysis been performed appropriately and rigorously? 

Reviewer #1: Yes

Reviewer #2: Yes

4. Have the authors made all data underlying the findings in their manuscript fully available?

Reviewer #1: Yes

Reviewer #2: Yes

5. Is the manuscript presented in an intelligible fashion and written in standard English?

Reviewer #1: No

Reviewer #2: Yes

6. Review Comments to the Author

Reviewer #1: Writing corrections:

A great many improvements have been made since the first version. There still are many more ways English usage needs to be improved, often due to revisions that were made. Some are hard for people not raised with English as the first language in their country (such as which preposition sounds right to native speakers based on customary use) and some involve the advanced rules used in professional writing in English (I use the Publication Manual of the American Psychological Association for guidance in writing for professional journals). In other places there was lack of clarity about what was meant.

Specific changes needed are listed.

Page 4: “composed by” should be “composed of”. Change “its likely to be beneficial in measuring craving in multifactorial scales, which can have” to “it is likely to be beneficial to use multifactorial scales for measuring craving, since these can have”. Change “is usually extensive and with a great number of items” to “usually has a great number of items”. Change “negative effect on respondents’ motivational and cognitive processes” to “negative effect on respondents’ willingness to accurately complete them”. Change “short-forms of multifactorial alcohol” to “short forms of multifactorial alcohol”. (The rule is to hyphenate the two words only when they together act as an adjective modifying another noun, according to my publication manual.)

Page 6 top: Change “non/low drinkers” to “non/low-risk drinkers”.

Table 1: It is necessary to have any abbreviations in a table explained in the table without people referring to the text, so you need a note defining M, SD. AUDIT, PACS, ACQ-SF-R.

Page 6: Measures: “is a self-reported measure” should be “is a self-report measure” (this is just the way it is phrased) for every measure you describe. Change “12–item” to “12 items” since “item” acts as a noun rather than as part of a modifying phrase (such as “47-item”) in this sentence. Change “Items are ranked on a” to “Items are rated on a”. (These have different meanings). Change “same four factors than ACQ-Now” to “same four factors as the ACQ-Now”. To meet the rules of parallel grammar, the following phrase must be changed from “the intent and planning to drink” either to “intending and planning to drink” or “the intention and plan to drink”.

Page 7: Top: Change “restrain capacity” to “restraint capacity” (restrain is a verb). Change “valitated to European Portuguese speakers” either to “validated in European Portuguese” or “validated for European Portuguese speakers”. At the end of the AUD section add: “This measure was translated into European Portuguese” or some similar statement.

Page 8, lower half: change “inter-items correlation” to “inter-item correlation” (since “inter-item” is being used as an adjective to modify a noun).

Page 9: Please fix this sentence by changing “Additionally, using the two groups based on alcohol use pattern (see Table 1), it was conducted a repeated measures ANOVA” to “Additionally, in order to compare ACQ-SF-R scores for participants in the two levels alcohol risk (see Table 1), a repeated measures ANOVA was conducted”. The last sentence on page 9 is unclear as to what was done; should “were assessed” be changed to “were compared between genders” or “were conducted separately within each gender”?

EFA Results: Since the Method section already said an EFA with Varimax was conducted, do not repeat that fact here. The Method did not say the EFA used ULS so you should add that to the Data Analysis Method. Change “comprised by items” to “comprised of items”. Change “double loaded to any factor” to “double loaded on any factor”.

Table 5 should be deleted. It is customary to take the authors’ word for the statement referenced and not waste space presenting those details.

Validity Results: Change “non/low drinkers” to “non/low risk drinkers” in two places. When significant differences are reported, the means and sds for each significant difference need to be presented. This could be done in a small table since you will present eight of each of these (total and each subscale).

Page 12, factorial invariance: What is meant by “groups by gender”? What groups? Should this say “When conducting EFAs separately within each gender”? What does “factor analysis by gender” mean? I have not heard of entering gender as a variable in a factor analysis. Was the EFA conducted this way? I think you meant to say item 1 had a higher factor loading “on” F2 than F3.

Page 13, top: “Likewise, mean inter-item correlation” should use the plural.

Discussion, 1st paragraph: “the ACQ-SF-R possesses good psychometric properties and reveals to be a valid and reliable measure” is not matching noun with verb correctly in that phrase. Instead it could say “the ACQ-SF-R possesses good psychometric properties and is a valid and reliable measure” (both verbs linking back to the measure as the subject of that clause). If “reveals” was supposed to refer to the sentence subject “findings”, then the wrong case was used, causing confusion about the referent. Change “showed cohesiveness between genders” to “showed cohesiveness within each gender”.

Bottom of page 13, third from last sentence “validation” should be plural, since referring to two other measures. Change “Thus, the present study seems to complement the PACS” to “Thus, the present study seems to complement the psychometric results found for the Portuguese version of the PACS”; then change “which to date is the only questionnaire validated in European Portuguese assessing alcohol craving” by moving “assessing alcohol craving” to after the word “questionnaire”, for clarity.

Page 14: Change “satisfactory concurrent validity with between ACQ-SF-R and AUDIT” to “satisfactory concurrent validity between the ACQ-SF-R and the AUDIT”. Change “non/low drinkers” to “non/low risk drinkers”. The following sentence is too strongly worded, given the lack of extant evidence supporting it: “Consequently, ACQ-SF-R seems to be a useful tool for clinical practice, helping to detect risky drinking, and predict relapses and treatment outcomes.” Change “seems to be” to “may be”, and eliminate mention of predicting relapse and treatment outcomes. You could say that future research could determine whether it will be useful in predicting risk for relapse and other treatment outcomes. These two terms are redundant: “four-factor dimensional measure” – remove either factor or dimensional.

Page 15: “reflecting the intent and planning to drink alcohol” is mixing verb with noun where parallel grammar is needed. Change either to “intending and planning to drink” or “the intention and plan to drink”. Change “and, consequently, low purposefulness to drink” to “and, consequently, scored low on the scale of purposefulness in regard to drinking”. (No one says “purposefulness to drink” in English so that is very hard to understand.) Change “Despite the sample size is higher” to “Despite the sample size being higher” or “Despite the fact that the sample size was higher”. Better yet, change “Despite the sample size is higher” to “While a strength of the study was that the sample size was higher”. Change “than previous validations of the scale” to “than previous validation studies of the scale”.

Page 16: Change “match the gender distribution” to “equate the number of each gender and include a general adult population”. The remove the phrase about greater sociodemographic representation to focus on the need for a clinical group in that next sentence. Change “it allows to assess” to “it allows assessment of”.

Reviewer #2: The authors have extensively responded to all my questions and accepted my suggestions. The manuscript has improved from the original submission.

7. PLOS authors have the option to publish the peer review history of their article (what does this mean?). If published, this will include your full peer review and any attached files.

Reviewer #1: No

Reviewer #2: No

---

## [Author Response · Author response to Decision Letter 1]

25 Apr 2021

We thank the reviewer for their useful suggestions. In the next pages, we try to address each of their considerations.

Reviewer 1

First of all, we thank the reviewer for their valuable comments and suggestions, specifically the care they gave into writing and grammar corrections and suggestions, as we believe they have helped substantially to improve the manuscript.

Issue 1. Writing corrections

1. On page 4, “composed by” was changed to “composed of”; “its likely to be beneficial in measuring craving in multifactorial scales, which can have” was changed to “it is likely to be beneficial to use multifactorial scales for measuring craving, since these can have”; “is usually extensive and with a great number of items” was changed to “usually has a great number of items”; “negative effect on respondents’ motivational and cognitive processes” was changed to “negative effect on respondents’ willingness to accurately complete them”; and “short-forms of multifactorial alcohol” was changed to “short forms of multifactorial alcohol”.

2. On page 6, “non/low drinkers” was changed to “non/low-risk drinkers”. This changed was also conducted in every instance the consumption group was named.

3. Further on page 6, “is a self-reported measure” was changed to “is a self-report measure”, and this correction was applied to every instance of the same error; “12–item” was changed to “12 items”: “Items are ranked on a” was changed to “Items are rated on a”; “same four factors than ACQ-Now” was changed to “same four factors as the ACQ-Now”; and “the intent and planning to drink” was changed to “the intention and plan to drink”.

4. On page 7, “restrain capacity” was changed to “restraint capacity”; and “validated to European Portuguese speakers” was changed to “validated for European Portuguese speakers”.

5. On page 8, “inter-items correlation” was changed to “inter-item correlation”.

6. On page 9, “Additionally, using the two groups based on alcohol use pattern (see Table 1), it was conducted a repeated measures ANOVA” was changed to “Additionally, in order to compare ACQ-SF-R scores for participants in the two levels alcohol risk (see Table 1), a repeated measures ANOVA was conducted”; and “were assessed” was clarified and changed to “were conducted separately within each gender”.

7. On page 10, “comprised by items” was changed to “comprised of items”; and “double loaded to any factor” was changed to “double loaded on any factor”.

8. On page 12, “When assessing groups by gender” was changed to “When conducting EFAs separately within each gender”; and “for F2 (Purposefulness) instead of F3” was changed to “on F2 (Purposefulness) than F3”.

9. On page 13, “Likewise, mean inter-item correlation” was written using its plural “Likewise, mean inter-item correlations”; “the ACQ-SF-R possesses good psychometric properties and reveals to be a valid and reliable measure” was changed to “the ACQ-SF-R possesses good psychometric properties and is a valid and reliable measure”; “showed cohesiveness between genders” was changed to “showed cohesiveness within each gender”; “validation” when referring to the previously validated versions of the scale was changed to “validations” since its plural form should be the one used; “Thus, the present study seems to complement the PACS” was changed to “Thus, the present study seems to complement the psychometric results found for the Portuguese version of the PACS”; and “which to date is the only questionnaire validated in European Portuguese assessing alcohol craving” was changed to “which to date is the only questionnaire assessing alcohol craving validated in European Portuguese”.

10. On page 14, “satisfactory concurrent validity between ACQ-SF-R and AUDIT” was changed to “satisfactory concurrent validity between the ACQ-SF-R and the AUDIT”; given its redundancy “dimensional” was removed from “four-factor dimensional measure”.

11. On page 15, “reflecting the intent and planning to drink alcohol” was changed to “reflecting the intention and plan to drink”; “and, consequently, low purposefulness to drink” was changed to “and, consequently, scored low on the scale of purposefulness in regard to drinking”; “Despite the sample size is higher” was changed to “While a strength of the study was that the sample size was higher”; and “than previous validations of the scale” was changed to “than previous validation studies of the scale”.

12. On page 16, “match the gender distribution” was changed to “equate the number of each gender and include a general adult population”, and the following sentence regarding greater sociodemographic representation was removed; “it allows to assess” was changed to “it allows assessment of”.

Issue 2. Table 1 was missing the necessary note defining the abbreviations used

We thank the reviewer for bringing out attention to this issue. The following note was added below Table 1 “Note: M, mean; SD, standard deviation; AUDIT, Alcohol Use Disorders Identification Test; PACS, Penn Alcohol Craving Scale; ACQ-SF-R, Alcohol Craving Questionnaire – Short Form – Revised.”; the remaining tables were checked for abbreviations that were not correctly described.

Issue 3. The AUDIT was not correctly described has being translated to European Portuguese

In agreement with the reviewer’s suggestion, we added the following sentence to our Measures on page 7 “In the present study, we used the European Portuguese translated version of the AUDIT”, hence, clarifying the scales translation into European Portuguese.

Issue 4. Information regarding EFA characteristics was mentioned in the Method and repeated in the Results

Following the reviewer’s suggestion, duplicated information describing the EFA was removed from the Results. Additionally, the sentence that mentioned the use of ULS in our EFA was moved so that it gets better highlighted during the reading of the manuscript. 

Issue 5. Table 5 removal

As highlighted by the reviewer, it is customary to take the authors’ word for possible changes in the reliability of the scale when removing any items. As such, Table 5 was considered by us redundant and consequently removed from the manuscript.

Issue 6. No information (means and standard deviation) regarding the significant differences in craving between the two consumption groups were reported

We would like to thank the reviewer for alerting us of this issue. Considering their suggestion, we added Table 5 to the manuscript following the Validity section of the manuscript.

Issue 7. Too strong wording was used when describing possible uses for the ACQ-SF-R

We sincerely thank the reviewer for highlighting this issue, and taking into account their suggestions. Thus, in the page 15, the phrase “Consequently, ACQ-SF-R seems to be a useful tool for clinical practice, helping to detect risky drinking, and predict relapses and treatment outcomes” was changed to “Consequently, ACQ-SF-R may be a useful tool for clinical practice, being able to help to detect risky drinking”

---

## [Decision Letter · Decision Letter 2]

3 May 2021

Portuguese Validation of the Alcohol Craving Questionnaire – Short Form – Revised

PONE-D-21-05321R2

Dear Dr. Crego,

We’re pleased to inform you that your manuscript has been judged scientifically suitable for publication and will be formally accepted for publication once it meets all outstanding technical requirements.

Kind regards,

Wen-Jun Tu

Academic Editor

PLOS ONE

Additional Editor Comments (optional):

Reviewers' comments:

Reviewer's Responses to Questions

**Comments to the Author**

1. If the authors have adequately addressed your comments raised in a previous round of review and you feel that this manuscript is now acceptable for publication, you may indicate that here to bypass the “Comments to the Author” section, enter your conflict of interest statement in the “Confidential to Editor” section, and submit your "Accept" recommendation.

Reviewer #1: All comments have been addressed

2. Is the manuscript technically sound, and do the data support the conclusions?

Reviewer #1: Yes

3. Has the statistical analysis been performed appropriately and rigorously? 

Reviewer #1: Yes

4. Have the authors made all data underlying the findings in their manuscript fully available?

Reviewer #1: Yes

5. Is the manuscript presented in an intelligible fashion and written in standard English?

Reviewer #1: Yes

6. Review Comments to the Author

Reviewer #1: (No Response)

7. PLOS authors have the option to publish the peer review history of their article (what does this mean?). If published, this will include your full peer review and any attached files.

Reviewer #1: No

---

## [Editor Report · Acceptance letter]

14 May 2021

PONE-D-21-05321R2 

Portuguese Validation of the Alcohol Craving Questionnaire – Short Form – Revised 

Dear Dr. Crego:

I'm pleased to inform you that your manuscript has been deemed suitable for publication in PLOS ONE. Congratulations! Your manuscript is now with our production department. 

Kind regards, 

on behalf of

Dr. Wen-Jun Tu 

Academic Editor

PLOS ONE